# Evaluating the sustainability and long-term outcomes of the Home Care Support Intervention Program (HoSIP) to reduce loneliness among community-dwelling older adults: A two-year follow-up study

Elham Lotfalinezhad[1,2,3], Shannon Freeman[4], Ahmad Kousha[2], Karen Andersen-Ranberg[5,6], Jeffrey E. Stokes[7], Sama Amirkhani-Ardeh[8], Ahmad Sohrabi[9], Farzaneh Barati[10], Mina Hashemiparast[11], Mohammad Asghari-Jafarabadi[12,13,14], Mohammad Reza Honarvar[15], Haidar Nadrian[16]*

**1** School of Nursing, University of Northern British Columbia, Prince George, British Columbia, Canada, **2** Department of Health Education and Promotion, Tabriz University of Medical Sciences, Tabriz, Iran, **3** Department of Psychiatry and Community Health Nursing, School of Nursing and Midwifery, Golestan University of Medical Sciences, Gorgan, Iran, **4** School of Nursing, Faculty of Human and Health Sciences, University of Northern British Columbia, Prince George, British Columbia, Canada, **5** Department of Clinical Research, University of Southern Denmark, Odense, Denmark, **6** Department of Geriatrics, Odense University Hospital, Odense, Denmark, **7** Department of Gerontology, University of Massachusetts Boston, Boston, Massachusetts, United States of America, **8** School of Health Sciences, University of Northern British Columbia, Prince George, British Columbia, Canada, **9** Cancer Control Research Center, Cancer Control Foundation, Iran University of Medical Sciences, Tehran, Iran, **10** Department of Surgery and Critical Care Nursing, School of Nursing and Midwifery, Golestan University of Medical Sciences, Gorgan, Iran, **11** Social Determinants of Health Research Center, Zanjan University of Medical Sciences, Zanjan, Iran, **12** Cabrini Research, Cabrini Health, Malvern, Victoria, Australia, **13** School of Public Health and Preventive Medicine, Monash University, Melbourne, Victoria, Australia, **14** Department of Psychiatry, School of Clinical Sciences, Monash University, Clayton, Victoria, Australia, **15** Health Management and Social Development Research Center, Golestan University of Medical Sciences, Gorgan, Iran, **16** Social Determinants of Health Research Center, Tabriz University of Medical Sciences, Tabriz, Iran

\* haidarnadrian@gmail.com

## Abstract

### Introduction

Understanding the long-term effects of home care support programs on loneliness in older adults is crucial for optimizing service delivery and improving the quality of life and care. This research explores the Sustainability and Long-Term Outcomes of the Home Care Support Intervention Program (HoSIP) to Reduce Loneliness among Community-dwelling Older Adults: A two-year follow-up study.

### Method and materials

This concurrent nested mixed-method study investigated the impact of HoSIP on older adults two years post-implementation. Quantitative data were collected on loneliness, social networks, perceived social support, quality of life, self-care ability,

**Data availability statement:** The data underlying this study are subject to ethical restrictions and cannot be shared publicly. Data access requests may be directed to the Research Ethics Committee of Tabriz University of Medical Sciences, Tabriz, Iran. Access to the data will be granted to qualified researchers upon reasonable request and subject to approval by the Ethics Committee, in accordance with institutional and ethical guidelines. Contact Information: • Email: research-vice@tbzmed.ac.ir • fax: +98 4133344280 • phone 1: +98 4133357310 • phone 2: +98 4133341249 • Website: https://researchvice-en.tbzmed.ac.ir/Page/26/Research-Ethics-Committee-%28REC%29.html?utm_source=chatgpt.com.

**Funding:** The author(s) received no specific funding for this work.

**Competing interests:** The authors have no conflict of interest to declare. This does not alter our adherence to PLOS ONE policies on sharing data and materials.

and general health. RAMNOVA analysis was used to analyze the results of univariate tests conducted at different points of measurement using SPSS version 23. Sixteen participants completed semi-structured individual interviews in-person and virtually. Conventional content analysis was undertaken using MAXQDA version 20.

## Results

Sixteen older adults remained in the HoSIP program at the two-year post-test assessment (mean age 73.5 years +6.6 years). The participants were predominantly female (81.3%). Over two years compared to baseline, a significant decline was observed in loneliness, social network, perceived social support, quality of life, self-care ability ($p < 0.05$) while no significant changes were observed for general health ($p > 0.05$). Three main categories, along with forth sub-categories, emerged from the data analysis.

## Discussion

This study explored how a community-based program helped reduce loneliness in older adults. The results highlight the importance of involving older adults in designing programs to improve their overall well-being. These findings can guide future interventions to enhance the quality of life for older adults, potentially lowering healthcare costs and benefiting both individuals and governments. This program provides a framework for the development and implementation of sustained, community-based interventions directed by older adults. Given the potential impacts of sociocultural factors on the efficacy and longevity of such programs, these elements warrant careful consideration during the design phase of the similar interventions.

## Introduction

Loneliness and social isolation among older adults are recognized as significant public health concerns, with both mental and physical well-being at risk [1,2]. Loneliness correlates with a range of adverse health outcomes, including increased risk of cardiovascular disease, depression, cognitive decline, and mortality [3,4]. As populations across the world, the challenges posed by loneliness among the older adult population are becoming more widespread, particularly in societies where traditional family structures are evolving. Urbanization and modern work demands may leave older adults without adequate social connections, which is increasingly seen in both Western and non-Western societies [5,6]. These global patterns are especially apparent in Iran, where shifts in population age, urban movement, and family dynamics have contributed to a larger older adult population at increased risk of social isolation [7,8]. Loneliness and social isolation in older adults not only increase vulnerability to depression and chronic disease but also place a substantial burden on healthcare systems through higher rates of hospitalization [9,10]. Addressing these issues is essential both for individual well-being and the sustainability of health services.

The role of social connectivity in promoting psychological well-being and cognitive health has become a focal point in geriatric and gerontological research. Engaging in social and community activities can be a protective factor for older adults from cognitive decline, supports independent living, and lowers institutionalization risks [11,12]. Through engagement in activities such as art, music, and technology, older adults can engage in communication with others, build skills, and help one another, especially with digital skills. Group discussions and support groups offer emotional support, while health and wellness activities provide practical guidance on managing health. These peer-based interactions enhance knowledge, social bonds, and cognitive well-being. These findings emphasize the need for older adults to alleviate loneliness and also to foster regular social interaction in environments where older adults can thrive. Studies have shown that older adults actively involved in community programs and engaged in meaningful activities contribute significantly to community cohesion and personal fulfillment; including volunteering, mentoring, or participating in social clubs [13,14]. Hands-on environmental efforts, such as planting along creek banks, protecting local habitats, planting trees, and gathering environmental data, were aimed at creating a better world for future generations, enhanced the wellbeing of older adults [14].

In Iran, like in other countries facing rapid demographic shifts, supporting the growing population of older adults has become a priority. Changes in family dynamics and increased urbanization have disrupted traditional support systems, leading to higher levels of isolation and loneliness among older Iranians [15–17]. Around the world, various interventions, including social facilitation interventions, psychological therapies, befriending interventions and leisure/skill development, have shown positive impacts in reducing loneliness and improving quality of life for older populations [18–20]. However, to be effective, it is important that community-based interventions be culturally adapted to meet the specific needs and context of Iranian older adults.

Challenges encountered by older adults, such as maintaining social engagement and managing health concerns over time, suggest a need for ongoing support and structured activities to sustain the intervention's benefits [5,19]. Furthermore, the program's success such as group-based exercises and internet-based interventions [21], illustrates the importance of accessible community spaces in fostering a sense of belonging and meaningful engagement among older adults [9,22,23]. For interventions to have sustained impact, they need iterative review, opportunity for adaptation and ongoing awareness and engagement reinforcement to meet the changing needs of older adults. In an enhanced fitness program, periodic updates like new exercises and intensity adjustments kept participants engaged and improved their physical health over time [24]. Similarly, a senior connections project showed that updating social activities to reflect participants' interests and cultures sustained connections and reduced loneliness among older adults [25,26]. These examples underscore the importance of flexibility and ongoing program development to ensure that interventions remain relevant and beneficial in the long term.

The present study examined the long-term effects the "Design, Implementation, and Evaluation of an Informal Home Care Support Intervention Program (HoSIP) for Lonely Older Adults in the Community", which was originally conducted from 2020 to 2022 [27]. The original study included three phases including a cross-sectional study which assessed the factors influencing subjective loneliness and quality of life among lonely older adults living in the community [28]. Following this, the insights gained from the initial study guided the design of HoSIP, implemented over a period of 12 weeks. The findings from this intervention were previously published [29]. This study aims evaluating the sustainability and long-term outcomes of the Home Care Support Intervention Program (HoSIP) to Reduce Loneliness among Community-dwelling Older Adults: A two-year follow-up study. Existing intervention studies targeting older adult populations typically concentrate their evaluation efforts on immediate post-intervention outcomes, failing to conduct longitudinal assessments of participants' subsequent experiences. Additionally, comprehensive evaluation of intervention effectiveness on target variables is rarely conducted during brief assessment windows (e.g., three months) or extended follow-up periods spanning 6–12 months. Consequently, implementing systematic evaluation of both sustainability and long-term intervention effects would serve a dual purpose: determining program durability while simultaneously measuring comprehensive intervention

impact across all study variables. Notably, the present intervention employs a participant-directed management approach, wherein older adults assume primary responsibility for program implementation without researcher interference, constituting a distinctive methodological innovation.

## Method and materials

### Study design

This mixed methods research employs a concurrent nested embedded design, prioritizing the quantitative component while also integrating qualitative data to provide a more comprehensive analysis.

### Participants and recruitment

The original HoSIP intervention was implemented over a 12-week period, from April 4 to July 3, 2021, with all program activities and participant engagement delivered within that defined timeframe. The first round of data collection—which included both quantitative and qualitative elements—was conducted shortly after, from July 4 to July 13, 2021, with an initial follow-up on July 14, 2021. The study then entered a post-intervention monitoring phase. The second round of quantitative data collection occurred from October 14 to October 19, 2021, followed by a second follow-up from October 20, 2021, to January 19, 2022. The third quantitative data collection was conducted from January 20 to January 24, 2022 [29], importantly, following the conclusion of the initial 12-week intervention, no further structured intervention activities were provided; rather, participants continued practicing the skills and behaviors learned during the original program independently. To assess the sustainability and long-term effects of HoSIP, an additional follow-up was performed two years after the last follow-up, from January 24 to February 16, 2024. This phase focused on evaluating the persistence of the intervention's impact on loneliness, quality of life, social engagement and overall health among the original participants.

### Data collection

As part of this ongoing mixed-methods evaluation, the researcher (EL) initially interviewed with older adults who exhibited the lowest levels of loneliness compared to their scores from a previous assessment conducted two years prior. Interviews were continued until data saturation was achieved—that is, until no new themes or insights emerged from the data. In this study, data saturation was reached after conducting interviews with a total of 16 participants. Meetings were held in public locations in Gorgan city, as well as in the participants homes or virtually as was preferred by the participant to administer questionnaires and conduct in-depth interviews. Validated instruments were used to measure key variables, including loneliness, social support, Perceived Social Support, General Health quality of life, and self-care ability. These instruments were:

- The questionnaires include the 20-item UCLA Loneliness Scale [30].

- The six-item Lubben Social Network Scale [31].

- The 12-item Multidimensional Scale of Perceived Social Support [32].

- The Control, Autonomy, Self-Realization and Pleasure Scale [33].

- The 12-item General Health Questionnaire [34].

- The 17-item Self-care Ability Scale for the Elderly [35].

Quantitative data were analyzed using statistical software SPSS version 23. The data were analyzed using descriptive statistics, frequency distribution tables, and a statistical test of Repeated Measures Analysis of Variance (RMANOVA), which was used to examine the univariate test at different measurement intervals for six variables of loneliness, general

health, and quality of life, self-care ability, perceived social support, and social networks. Post hoc Bonferroni test was also applied to investigate the long-term effect of HoSIP on outcome variables.

Individual interviews were thoroughly conducted using an interview guide through face-to-face and online sessions, with each session averaging 40 minutes. The interview guide was constructed based on Social Support Theory, which highlights the crucial importance of social connections in affecting health and wellness in aging populations. This theoretical approach guided our examination of anticipated long-term effects, the advantages and challenges encountered by participants, and how newcomers influenced the group dynamics within the HoSIP program. Through anchoring our inquiries in Social Support Theory, we sought to document both personal and group experiences generated through engagement in peer support programs. Conventional content analysis and semi-structured interviews were used to collect and analyze the qualitative data. Three questions of focus in the interviews included: 1-As a participant of the program, what influences may you have perceived as the long-term outcomes of HoSIP on your life? 2- What is your idea about any possible benefits and/or barriers of the program for the newly-joined older adults to the HAMDAM group? 3- What positive and/or negative things may newly-joined older adults added to this program? To explore the depth of participants' experience, probing questions were used where needed including "why", "how", and "could you please explain this a little more". The design and reporting of the qualitative component were guided by the Consolidated Criteria for Reporting Qualitative Research (COREQ) checklist to ensure transparency and rigor [36], guided the semi-structured interviews. The qualitative data were analyzed using the qualitative data analysis software MAXQDA version 20.

Audio recordings were transcribed into verbatim. Each interview was read for several times to gain a comprehensive sense of participants' thoughts. Meaning units were extracted from participants' statements, which were then condensed into codes. Categories and subcategories were developed using an inductive (conventional) content analysis approach, whereby all categories and subcategories emerged directly from the data rather than from predefined frameworks. This ensured that the identified themes were grounded in the actual responses of the participants. The authors (EL, HN, and MH) checked back and forth the extracted codes, subcategories and categories to compare similarities and differences. Furthermore, in the research team member-checking and analytical triangulation were evaluated for emerging subcategories, categories and quotes. To ensure trustworthiness of the study, the feedback of the interviewer and interviewee was done. The findings of thematic analysis were reviewed and approved by the research team and all authors agreed upon the final version. Written informed consent was signed by all participants. Ethical approval of this study was granted by Tabriz University of Medical Sciences, Tabriz, Iran (IR.TBZMED.REC.1399.488).

## Results

Out of the 36 older adults who originally participated in HoSIP, 12 withdrew for personal reasons, and 8 others remained connected only through the WhatsApp group, declining to participate in any in-person sessions, 16 (n = male: 3, and female:13; 44.5%) remained in the follow-up phase and actively engaged with HoSIP, both online and in-person were contacted for this study. The mean age of participants was $73.50 \pm 6.58$, of whom 81.3% were women. Level of education ranged among participants with 43.8% reporting a secondary or tertiary education (n = 7). Approximately half of participants received an old-age pension (n = 9). Table 1 shows the demographic characteristic of the participants.

Table 2 indicates the results of Repeated Measures Analysis of Variance (RMANOVA) at different measurement intervals for six variables of loneliness, general health, and quality of life, self-care ability, perceived social support, and social networks. The results demonstrated statistically significant changes from baseline to two-year follow-up for felling of loneliness, quality of life, perceived social support, social network and self-care ability ($p < 0.05$). However, there was not a significant difference from baseline and 2 years for general health ($p > 0.05$). All variables except for general health, had a significant difference over these two years and these changes were in the direction of improvement (Fig 1). The results of post hoc Bonferroni test to investigate the long-term effect of HoSIP illustrated that except for UCALA, all other variables were close to their original state and the changes were not significant.

**Table 1. Participant demographics (N = 16).**

| Participant Characteristic | N | % |
|---|---|---|
| **Age (years)** | | |
| Young old (60–74) | 12 | 75 |
| Old-old (75–84) | 3 | 18.8 |
| Oldest-old (≥85) | 1 | 6.3 |
| **Gender** | | |
| Men | 3 | 18.8 |
| Women | 13 | 81.3 |
| **Education** | | |
| Illiterate | 2 | 12.5 |
| Primary education | 7 | 43.8 |
| Secondary education | 6 | 37.5 |
| Tertiary education | 1 | 6.3 |
| **Occupation** | | |
| Retirement | 4 | 25 |
| Pensioner | 9 | 56.3 |
| Household | 2 | 12.5 |
| Part-time job | 1 | 6.3 |

**Table 2. Repeated measure analysis examining the univariate test at different measurement intervals.**

| Variables | Univariate Tests | |
|---|---|---|
| | **Sphericity Assumed** | **Greenhouse-Geisser** |
| **UCLA**[1*] | | < 0.001 |
| **CASP**[2*] | < 0.001 | |
| **GHQ**[3*] | 0.358 | |
| **MSPSS**[4*] | 0.009 | |
| **LSNS**[5*] | 0.007 | |
| **SASE**[6*] | | 0.037 |

[1*]UCLA, UCLA loneliness;

[2*]CASP, Control, Autonomy, Self-realization, and Pleasure;

[3*]MSPSS, Multidimensional Scale of Perceived Social Support;

[4*]LSNS, Lubben Social Network Scale;

[5*]SASE, Self-care Ability Scale for the Elderly;

[6*]GHQ12, General Health Questionnaire.

All participants who remained in the HoSIP after two years agreed to participate in the study (N = 16). The gender distribution among the interviewees was 81% women and 19% men. Overall, 150 initial codes were extracted, which were summarized in 3 main categories and 14 sub-categories. The thematic analysis led to the identification of the long-term effects of HoSIP, as shown in Table 3.

## 1. Interpersonal relationships

This category showed the favorable lasting impacts of the HoSIP program, which contains seven subset theme areas as described below:

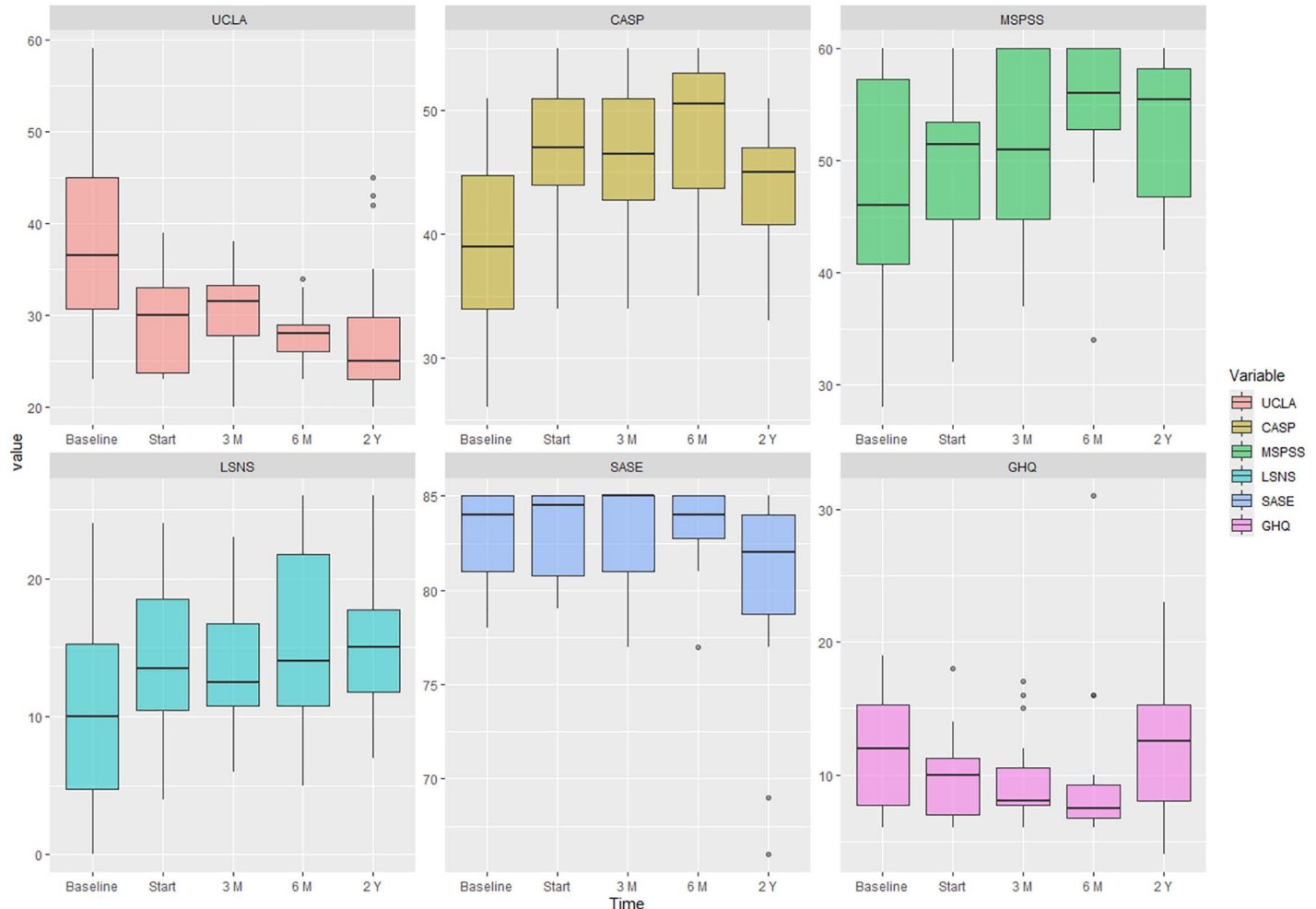

**Fig 1. The results of Repeated measure analysis of variance from baseline to Post-Project Assessment (2 years, 2022-2024).**

**Table 3. Extracted themes and sub-themes.**

| Themes | Sub-themes |
|---|---|
| **Interpersonal relationships** | Finding friends and companionship<br>Understanding each other<br>Helping each other<br>Giving motivation and encouragement to each other<br>Learning from each other<br>Trusting each other<br>Improving the sense of intimacy |
| **Personal growth and well-being** | Paying attention to themselves<br>Becoming more sociable<br>Feeling useful<br>Decreasing use on pharmaceutical interventions<br>Improving self-confidence |
| **Mood and behavior** | Positive changes in mood<br>Changing attitudes and behavior |

**1.1. Finding friends and companionship.** Many respondents highlighted the value of forming friendships and finding camaraderie through involvement in the HoSIP program. As one respondent expressed:

*"I have found more friends in addition to the friends I already had. This is much better because in our age, what benefits me the most is having friends that fill my loneliness and emptiness." (participant 14, female, aged 74).*

Through the HoSIP program, two lonely older adults got married. Prior to the marriage, the older man who participated in the HAMDAM group did not have any social events or entertainment in his life.

*"Before it was like I was an orphan but now this lady is by my side. We go on trips together. Since 2005 when my wife has passed way, I have not gone to Mashhad (holy shrine), but we go to Mashhad twice a year now" (participant 5, male, aged 69).*

**1.2. Understanding each other.** Older adults require acceptance and understanding from others due to psychological changes that come with age. They may not be understood by close family members, but older people tend to find acceptance among their peers who are in similar living situations and can relate to their experiences.

*"Maybe our children will never understand us the way we understand each other. If we travel or do anything, and I get tired and say, "oh my back hurts", my friends who are like me will also say their back hurts too. But my child doesn't have that problem so it's likely to be annoying for them, but these friends we share a common pain and perspective" (participant 4, female, aged 7).*

*"I think our empathy has increased; our group has become warmer; we are not like some groups that fall apart. we've become more united. Look, everyone has some flaws, but we only see the good things in each other" (participant 14, female, aged 74).*

**1.3. Helping each other.** Several respondents pointed to the advantage of having a peer support system. For example, they help one another out whenever someone needs assistance or encouragement from the group.

*"Mr. Kh, who's just joined, is a very respectable man, he's great. When he wants to go home, he says "I am leaving, and I'll give you a ride". he even drops me off to the end of my alley. Sometimes he asks me "do you need to go stopping on the way?" It's really nice... Very good... Now he gives a ride to 2-3 friends". (participant 14, female, aged 74).*

*"The fact that I can tell the head of HAMDAM group (Mr. Mh, Mrs Ch) if I'm lacking something. I share my troubles with Mrs N as a friend. if they can, they help me even financially, they help me with a loan that I'll repay, I can say they've become part of my family" (participant 10, female, aged 71).*

**1.4. Giving motivation and encouragement to each other.** The respondents indicated that they were able to gain positive energy from interacting with one another through the HoSIP program. Supporting quotes from participants further highlighted this viewpoint:

*"I asked the newcomers, and they said "it's a great place and we wish we had come here earlier. This group makes us feel better and gives us energy." (participant 15, female, aged70).*

*"The presence of new people who recite poetry, sing songs, and talk gives us energy." (Participant 16, female, aged 79).*

**1.5. Learning from each other.** Some participants shared that by taking part in the intervention program, they were able to learn many new things (such as poetry, art & crafts) from their peers. For example, participants made remarks such as:

*"We tried having more friendly meetups at members' homes, like cooking, knitting classes, I teach knitting at my café."* (Participant 6, female, aged 67).

*"It's interesting that I learned not to expect anything from anyone, not even from my children... Yes, I learned not to expect anything from anyone and to lower my expectations. For example, when my children don't call me today, I don't care, and I do not think that they forgot about me. I say, "Oh, they didn't make time, they didn't have the opportunity." They will definitely call me tomorrow. And that's what happens. I have a son who even calls me in bed when he's about to sleep. He says, "Mom, I'm sorry, I didn't get a chance to call you today. Are you okay? Everything is fine."* (participant 7, female, aged 69).

*"I have learned poetry. There was one poem that Mrs Najar says "If that Turk of Shiraz gets it, I will give our hearts to his Hindu mole, Samarkand and Bukhara" [poem in Persian].* (Participant 9, female, aged 79(.

**1.6. Trusting each other.** The participants observed that long-lasting impacts of the HoSIP could include fortifying the sense of community and togetherness, as well as enabling financial contributions over time.

*"Their trust has increased a lot before they would comment on everything they heard. they didn't think about whether what someone said was good or bad. They quickly judged and commented. Sometimes, it would make the group tense. But now, thank God, many of the members have gotten better and they don't talk about that at all. Fortunately, if, God forbid, someone has a problem, they call me. So, when they talk to me, it means they trust me."* (participant 1, female, aged 64).

*"This community that has now formed with this group has increased the group's trust in me. We opened a fund called the Qarz al-Hasanah Fund. They give us 100, 200, 300 hezar tomans [Iranian currency] every month as a fund that we keep in the fund. We told them that you can withdraw the money you give whenever you want, but if it remains, you will not get any profit. What do we do with the money that we get as profit? We spend it for our members. We take them on ecotourism... We take them walking. All these things that I am mentioning are under one condition, but they accept one person. Some people accept one person's group and trust them, and they trust them."* (Participant 2, male, aged 79).

**1.7. Improving the sense of intimacy.** Participants emphasized the value of a strong sense of intimacy and shared identity within the group. The close-knit atmosphere among the members has resulted in deep bonds being formed through the HoSIP program.

*"We have a common trait and the sense of intimacy. if you go somewhere else, you might not see this sense of intimacy. Everyone who joins the group has that sense of intimacy and being of the same type, especially meaning that we all think we are in the same category and are the same."* (Participant 1, female, aged64).

*"At first, it was hard for me to accept, but then I gradually got used to it, so that now, even though I go less often than some people, I see that I have to go. I mean, I see in myself that I have to go, not just saying that I need to go there, but I see in myself that I have to go there to see my friends. To see my loved ones, because for me, they have become really dear people who I can love. They are no different from my loved ones in terms of my own body, and my blood. I also developed a certain attitude towards them."* (Participant 6, female, aged 67).

## 2. Personal growth and well-being

This overarching category encompassed five sub-theme categories that described the personal impacts of participating in the HoSIP program.

### 2.1. Paying attention to themselves.

Lonely older adults stated that through participating in this program, their sense of self-awareness and attention to themselves has improved. This was reflected in the statements of some participants:

*"We can love ourselves more than before. Now when I do something good, I invite myself out and throw a birthday party for myself. This means you love yourself. Here I realized this is self-love"* (participant 7, female, aged 69).

*"When I want to come and participate in this session, I prepare myself as much as I can. And in the community, my speech should be such that I am not shamed, and I should be attractive, not repulsive"* (participant 2, male, aged 79).

### 2.2. Becoming more sociable.

Older adults explained that their social interactions have improved through the program. Some participants mentioned:

*"Some older adults were very shy at first. they wouldn't come up and introduce themselves or wouldn't say anything. But now they have changed in a good way."* (participant 14, female, aged 74).

*"I think many older participants were like me who the first days and they couldn't speak at all, I mean their hands and feet would shake while speaking, but now here they can present in the front of the audience and speak and recite poems very comfortably"* (participant 1, female, aged 64).

### 2.3. Feeling useful.

Most participants expressed that they were able to share their skills with others in the program, which led to positive feelings for them since it helped others. Two participants positively commented:

*"I'm very happy to see that when I speak, for example I summarize some texts from various books like psychology or anything. I see they welcome it and like it, and it gives me that feeling of being useful. It's a good feeling"* (participant 4, female, aged 71).

At our age, everyone thinks we've become passive, useless and idle, while we can still teach many skills like our life skills to even younger than us. Now when I look at my friends, I see they have such beautiful voices" (participant 1, female, aged 64).

### 2.4. Decreasing use on pharmaceutical interventions.

The respondents reflected that medical expenditures associated with treating various diseases could decrease if these kinds of community groups were created more widely. Participants made the following statements related to this:

*"This group should increase... Their numbers should increase nationwide, and this increase will reduce many of the country's expenses. You know, my concern has decreased in the two years when I've been here. I haven't gone to the doctor anymore, I mean, I haven't taken someone's place, I haven't taken medicine. So, who is all this for? For the benefit of the government? That is; by establishing different groups of older adults, we reduce extra expenditures."* (participant 2, male, aged 79).

*"Well, this is the positive aspect of this group. when women are happy, cheerful, and sociable; they are going out; they are not confined to their homes and are not isolated, this is a great help to the government and society because it leads to consuming less medicine and reducing the costs of treatments."* (P12) (participant 12, female, aged 74).

**2.5. Improving self-confidence.** Many respondents stated that participating in the group significantly boosted self-confidence among lonely older adults, enabling them to communicate more effectively in large groups. Two participants shared:

*"My self-confidence has increased a lot. Before, I couldn't speak in public at all. I couldn't communicate with people around me, and I was isolated. But now, my self-confidence has increased so much that I can communicate with most people. I can communicate verbally... Well, look, of course, when my self-confidence increases here, I can talk to anyone, in a group of 40, 50, and 60 people. when I go out, I can talk easily in any group. It's like it's infiltrated me, and I don't want to lose that anymore."* (Participant 1, female, aged 64(.

*Through participating in this group, our self-confidence has increased, we can do our own personal routines."* (Participant 7, female, aged 74).

## 3. Mood and behavior

This category consists of two sub-themes: Positive changes in mood and changing attitudes and behavior.

**3.1. Positive changes in mood.** Lonely older adults were able to enhance the vibrancy and vitality in their lives through taking part in this program. Participants made the following statements related to this:

*"Occasionally, if possible, I go on a trip. When I'm at home, I have to think, but when I come here, it's very good. I'm satisfied and happy, and my mood has changed. I sit at home and think all sorts of thoughts and feel sad, but when I come there, I'm happy."* (participant 13, female, aged 69).

*"A woman whose husband had died was in a bad mood. Since she came to us, she has been writing poetry, writing texts, reciting poems, and reading books. So, these are the positive aspects of our class. For example, she was very depressed and didn't talk at all, but now she is participating in discussions"* (participant 3, female, aged 71).

**3.2. Changing attitudes and behavior.** Several respondents related to the experience of lonely older adults changing their attitudes and behaviors through the program

*"we've gotten better, there are many of us now. Women have gotten better. For example, in terms of behavior, we didn't know each other well at first, but now that we have this same attitude".* (P9) (participant 9, female, aged 79).

*"A woman has recently joined the group. At first, she was very indifferent, but now her clothes have changed. she wears bright colors. she wants to talk all the time. If she doesn't talk, she gets angry and says, "Why don't you give me a turn to talk?" Before she was quiet and calm. In all this group only one person caught my attention was her. Her clothes have changed. Her way of talking has changed. It's as if her appearance and appearance have changed."* (Participant 7, female, aged 69)

## Discussion

The Home care Support Intervention Program (HoSIP) is a peer-to-peer support initiative designed to address loneliness among older adults. To assess the sustainability and long-term effects of this program, an additional follow-up was conducted two years after the intervention period. The findings indicate that HoSIP not only effectively improved the physical and psychosocial health of lonely older adults but also led to enhancements in their social networks and perceived social support.

Making new friends and fostering long-lasting relationship with peers positively influenced the older adults' lives, providing them meaning, value, and purpose. This aligns with previous research demonstrating that older adults benefit from higher life expectancy, companionship, identity, emotional supports based on functional theory in aging [37], considering digital technology integration as offering older adults' new meaningful roles while risking exclusion through unequal access, explaining both societal and individual impacts of digital transformation on aging populations [38]. Our finding support this, showing the positive outcomes from interpersonal relationships including receiving energy from each other, learning from friends, establishing relationships with new friends and companions, building trusting with others, enhancing understanding of the needs, feelings and values of others and helping others expanded participants social network and improved their perceived social support. These outcomes expanded participants' social network and improved their perceived social support, thereby promoting social capital in aging. According to Simons and et al, staying connected through digital media in old age can improve social capital, enhance mental health by reducing depression and loneliness in older adults [39]. This finding aligns with sustained loneliness improvement observed in our two-years follow-up study of HoSIP.

In addition to social benefits, participants experienced improvements in quality of life—particularly in psychosocial aspects such as mood, behaviour, and self-confidence. Since joining the group, some older adults reported greater self-expression and the chance to engage in meaningful volunteer activities, including helping people with disabilities, supporting underprivileged children, learning French, and taking part in cognitive training exercises [40]. These results support finding of a systematic review, in which also noted that improving self-confidence can have a predominant role in the performance of the athletics. Furthermore, participation in HoSIP resulted in participants paying more attention to their bodies, physical functioning, and their feelings, corroborating the ideas of Harleah et al, who suggested that older patients with higher self-care confidence tended to have better overall health and well-being in various aspects of their lives [41]. Post-project, the HAMDAM group members have shown remarkable initiative in organizing a variety of recreational, educational and cognitive activities. They have become more interconnected, more expressive in their community, and more knowledgeable about their rights. These factors have spurred them to independently arrange trips to various cities across Iran and to seek further education on topics relevant to aging and even recreational activities. It is significant that older adults' involvement in arranging activities could have such a positive impact on their psychological well-being [42]. Based on researcher's observations some older adults could not adapt to a newly arrived older adults and felt connected to people had been in the group since first years of starting project due to similar living and social circumstances. This experience is in line with Jeroense et al. study, showing some attributes including age, gender, and marital status may influence interpersonal dynamics [43]. Furthermore, the group initially used a daycare center for their activities, contributing to its upkeep through rental payments. However, after a year, the group relocated to a new location. This move resulted in spatial limitations, forcing the cancellation of some classes and most older adults were not able to attend their weekly face-to-face meetings because the new location was not accessible to older adults and required additional travel time, energy, and costs for commuting, reducing some social connection with their peers. Therefore, some older adults transitioned to rely on engagement with the group through the online social network. The limited space of new location led to further barriers for conducting some physical activities including volleyball and football. These barriers could have negative influence on their overall mental health after two years. According to aging in place theory, relocation stress in old age can lead to depression, anger and powerless [43], because older participants had gotten used to that place and changing the place was far painful for them. The new meeting location did not provide older adults a wide space for doing some entertainment activities such as playing volleyball and doing exercise. Prior studies confirmed earlier findings showing that contribution in ongoing physical activities and sports have an important influence on the emotional and physical health of lonely older adults [44].

A key strength of the present study was the demonstrated sustainability and organic growth of the intervention beyond the formal study period. The older adults remained engaged in the HoSIP and expanded their membership by inviting older adults living with spouses or children to join. As a result, membership increased over the past two years, with new

individuals joining the group. At the time of the two-year follow-up, 93 older adults were participating in the online and face-to-face HAMDAM group, of whom 57 had joined during the follow-up period since the program formally ended. Membership continued to rise as participants invited friends and relatives to join. Additionally, seasonal face-to-face talks on topics such as healthy aging, nutrition, and physical activity were introduced during the follow-up period. Every two months, older adults themselves invited experts from various fields to these in-person meetings, leading to empowering, participant-led educational sessions that helped build the older adults' autonomy.

Several important limitations need to be considered. First, the small sample size was an important limitation of this study. Eight older adults who had been active in the online social network were unable to attend the planned face-to-face group meetings in their homes or communities, which limited our ability to fully assess the HoSIP intervention's long-term effects on emotional, instrumental, informational, and affiliational support for these individuals. In addition, during the follow-up period, a change in the location of group gatherings—beyond the control and preferences of some older adults—led to further participant withdrawals, reducing the sample size even more. These factors combined to decrease the number of original participants available for recruitment at the two-year follow-up. The final sample size (n = 16) may have reduced the statistical power of our analyses and limited the generalizability of the findings. Despite this limitation, our findings offer valuable insights into the long-term impacts and challenges of sustaining community-based interventions for older adults. Second, the lack of a control group constrains researchers' capacity to establish causal relationships regarding the intervention's impact. In the absence of a comparison group that did not undergo the intervention, determining whether documented changes result from the intervention itself or from other variables—including temporal variations in outcomes or external factors—becomes challenging. Third, this study's qualitative findings did not include any negative themes or critical perspectives. This may be influenced by factors such as social desirability bias, group dynamics, or the voluntary nature of participation, which could have led participants to emphasize positive experiences and downplay or withhold negative feedback. As a result, the findings may not fully capture the range of participant perspectives or potential challenges associated with the long-term effects and sustainability of the HoSIP program.

These findings provide the following insights for future research: 1- In addition to regular face-to-face meetings, older adult should be encouraged where possible to also engage in the group remotely. This will allow for increased access to the program as well as a range of opportunities for further engagement. As technology becomes more integrated into the daily lives of older adults across the world [45], and the use of and familiarity with use of information communication technology such as computers, tablets, and smartphones increased, the numbers of older adults confident and confident to use technology will also increase. Since some participants were assigned responsibility for managing other participants, there were complaints and problems. 2- Any intervention program should provide clear written principles and rules of conduct so that all participants are aware of expectations for the group and can respect the established guidelines. 3- Additional studies using the HoSIP method among various ethnic, cultural, and religious groups would be valuable to give a better understanding of factors influencing loneliness, quality of life, and other related outcomes in these diverse populations. 4- Given that Iran's Ministry of Health is educating master's and Ph.D. students in aging studies, there is a need to create employment opportunities for these graduates. To address this, the government should integrate cost-effective community-based programs for older adults into healthcare services. This could involve establishing informal homecare support agencies, allocating funds for their operation, and allowing aging students graduates to supervise these centers. Importantly, the Ministry of Health must implement a registry and safety measures to ensure the well-being of older participants and students in these types of community-based programs.

## Conclusion

Initial benefits experienced by participants in the HoSIP program were sustained over time at two-year post-program follow-up. The interventions, co-created with lonely older individuals, positively impacted older adult participants social, emotional, and physical well-being. This project provides valuable insights for future interventions aimed at improving the

quality of life for older adults in various care settings. These insights can help reduce the costs associated with hospital readmissions and long-term treatments by implementing simple and cost-effective community-based programs. Governments can benefit from these free and easily accessible resources to implement innovative interventions for community-dwelling older adults. However, it's crucial for governments to oversee these activities within their communities to ensure safety and legal compliance. This model could serve as a blueprint for aging policies and community health strategies in other middle-income countries.

## Acknowledgments

We would like to thank the participants and the authorities of the National Pension Fund (Kaneh Mehr), the daycare authorities in Gorgan City and Aliabad-E Katul County (Kanoon Salmandi Jahandi Degan and Yas), without whose help this work would never have been possible.

## Author contributions

**Conceptualization:** Elham Lotfalinezhad, Shannon Freeman, Ahmad Kousha, Karen Andersen-Ranberg, Mina Hashemiparast, Mohammad Reza Honarvar, Haidar Nadrian.

**Data curation:** Elham Lotfalinezhad, Ahmad Sohrabi, Mohammad Asghari-Jafarabadi, Haidar Nadrian.

**Formal analysis:** Elham Lotfalinezhad, Sama Amirkhani-Ardeh, Ahmad Sohrabi, Mohammad Asghari-Jafarabadi.

**Investigation:** Elham Lotfalinezhad.

**Methodology:** Elham Lotfalinezhad, Shannon Freeman, Ahmad Kousha, Mohammad Reza Honarvar, Haidar Nadrian.

**Project administration:** Elham Lotfalinezhad, Karen Andersen-Ranberg.

**Software:** elham lotfalinezhad, Ahmad Sohrabi, Mina Hashemiparast, Mohammad Asghari-Jafarabadi.

**Supervision:** Elham Lotfalinezhad, Ahmad Kousha, Karen Andersen-Ranberg, Jeffrey E. Stokes, Mina Hashemiparast, Haidar Nadrian.

**Validation:** Elham Lotfalinezhad, Farzaneh Barati, Haidar Nadrian.

**Visualization:** Elham Lotfalinezhad.

**Writing – original draft:** Elham Lotfalinezhad, Sama Amirkhani-Ardeh, Farzaneh Barati.

**Writing – review & editing:** Elham Lotfalinezhad, Shannon Freeman, Ahmad Kousha, Karen Andersen-Ranberg, Jeffrey E. Stokes, Ahmad Sohrabi, Farzaneh Barati, Mina Hashemiparast, Mohammad Asghari-Jafarabadi, Mohammad Reza Honarvar, Haidar Nadrian.

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
