## [Decision Letter · Decision Letter 0]

18 Jul 2025

Dear Dr. Nadrian,

Thank you for submitting your manuscript to PLOS ONE. After careful consideration, we feel that it has merit but does not fully meet PLOS ONE’s publication criteria as it currently stands. Therefore, we invite you to submit a revised version of the manuscript that addresses the points raised during the review process.

We look forward to receiving your revised manuscript.

Kind regards,

Mehdi Rezaei

Academic Editor

PLOS ONE

Journal Requirements:

2. Please describe in your methods section how capacity to provide consent was determined for the participants in this study. Please also state whether your ethics committee or IRB approved this consent procedure. If you did not assess capacity to consent please briefly outline why this was not necessary in this case.

3. Thank you for stating the following in the Competing Interests section: There is no conflict of interest

4. Please note that your Data Availability Statement is currently missing the repository name and/or the DOI/accession number of each dataset OR a direct link to access each database. If your manuscript is accepted for publication, you will be asked to provide these details on a very short timeline. We therefore suggest that you provide this information now, though we will not hold up the peer review process if you are unable.

Reviewers' comments:

Reviewer's Responses to Questions

**Comments to the Author**

1. Is the manuscript technically sound, and do the data support the conclusions?

Reviewer #1: Yes

Reviewer #2: Partly

2. Has the statistical analysis been performed appropriately and rigorously?

Reviewer #1: Yes

Reviewer #2: No

3. Have the authors made all data underlying the findings in their manuscript fully available?

Reviewer #1: Yes

Reviewer #2: Yes

4. Is the manuscript presented in an intelligible fashion and written in standard English?

Reviewer #1: Yes

Reviewer #2: Yes

Reviewer #1: Dear editor Dr. Mehdi Rezaei Academic Editor in PLOS ONE

Thank you for your invitation to review manuscript entitled “The Effectiveness of the Home Care Support Intervention Program (HoSIP) to Reduce Loneliness among Community-dwelling Older Adults: A two-year follow-up study”

This article offers valuable and timely insights into the long-term effects of a community-based peer support intervention (HoSIP) aimed at reducing loneliness and improving the quality of life among older adults. One of the main strengths of this study is its mixed-methods design, which combines quantitative measures with in-depth qualitative interviews to provide a comprehensive understanding of both outcomes and lived experiences.

The study is particularly significant because it follows participants for two years after the intervention, capturing sustained impacts on social connectedness, self-confidence, and emotional well-being. This long-term perspective is often lacking in similar research. Moreover, the article highlights the adaptability and sustainability of the intervention, showing how older adults continued to expand and self-manage the program beyond the initial implementation.

Introduction

Comment: Consider breaking this introduction into shorter paragraphs to improve readability and emphasize key ideas, such as the health risks of loneliness, societal trends, and intervention strategies.

Comment: The number of citations in the first paragraph is quite dense. Consider integrating the references more smoothly to avoid overwhelming the reader and to improve narrative flow.

Comment: Strengthen the transition between global trends and the specific case of Iran. For example, a sentence explicitly linking the global phenomenon to Iran’s demographic shifts would improve the logical continuity.

Comment: While the background is comprehensive, clarify earlier why a long-term follow-up of the HoSIP program is necessary and what gap it addresses in current literature.

Comment: When referencing "various interventions" globally, it would be helpful to briefly name a few types (e.g., befriending programs, telehealth-based support) to give readers a clearer idea of what has been tried.

Comment: The final paragraph introduces your study, but the specific research question or hypothesis is not fully clear. Consider adding a sentence like: “This study aims to evaluate whether the effects of the HoSIP program are sustained two years after completion.”

Comment: The phrase “older adults” is consistently used, which is good. Ensure the same terminology is used throughout the paper, and avoid switching to "elderly" or “seniors” unless quoting from other sources.

Method

Comment: The phrase “This mixed method study with a concurrent nested embedded design” is technically accurate, but could be clarified for broader readability. Consider briefly describing what is "embedded" in what — e.g., “a primarily quantitative study with a qualitative component embedded for deeper exploration.”

Comment: The recruitment period is described with the exact dates. Consider aligning this level of specificity with the rest of the timeline (e.g., start and end dates of the intervention) for consistency and clarity.

Comment: The sentence “16 participants (Of the 36 lonely older adults...)” needs refinement in grammar and structure. Consider rephrasing as: “Out of the 36 older adults who originally participated in HoSIP, 16 remained in the follow-up phase and were contacted for this study.”

Comment: The long list of scales makes the paragraph heavy. You could introduce them with a sentence like: “Validated instruments were used to measure key variables, including loneliness, social support, psychological well-being, and self-care ability.” Then list them in smoother format, possibly in a bullet point or table format in the appendix.

Comment: Consider briefly explaining how the interview guide was developed. Was it based on previous literature, theoretical models, or pilot testing?

Comment: You could clarify how categories and subcategories were developed — was it inductive or deductive content analysis? That would help readers understand your analytic strategy.

Comment: Mentioning COREQ is excellent. Consider explicitly stating that the checklist guided both the design and reporting of the qualitative component for transparency.

Discussion

Comment: The opening sentence of the Discussion is long and slightly unclear. Consider breaking it into two: one describing the HoSIP program and another summarizing its long-term effects more clearly.

Comment: Good integration of functional theory and digital aging. Consider making the theoretical linkages even more explicit — e.g., how does this support or extend the theory, not just echo it?

Comment: The paragraph discussing quality of life and self-confidence would be stronger if you clearly stated the transition: e.g., “In addition to social benefits, psychological improvements such as self-confidence were also reported.”

Comment: In the paragraph starting “These findings provide the following insights…”, consider numbering or bulleting the insights (1, 2, 3, etc.) for clarity and easier reading. Each insight is important but gets lost in the long paragraph.

Comment: The Conclusion nicely summarizes the findings. You could enhance it by adding: “This model could serve as a blueprint for aging policies and community health strategies in other middle-income countries.”

Reviewer #2: This paper makes a meaningful contribution to the field, though some methodological concerns should be addressed.

Introduction

-This manuscript aims to evaluate the long-term effects of the HoSIP intervention and serves as a follow-up to the authors’ previous publication. The purpose of the study is not well stated. Was the study aimed at assessing the sustainability of the intervention after it was discontinued after two years? Or is the intervention still ongoing?

Method

-While the manuscript is well-referenced, the Methods section lacks sufficient detail. In particular, the description of the intervention is unclear. As, mentioned, it is not specified whether the intervention is ongoing or if it was limited to the initial 12-week period. Clarification is needed on whether this study is assessing the sustainability of the intervention's effects two years after its conclusion, or if the intervention has continued in some form during that time. Since the stated aim of the study is to examine the sustainability of the intervention, a more explicit explanation of the timeline and implementation status is essential.

-Additionally, the data analysis section requires further elaboration. Although the Results section lists the statistical tests used, it does not clearly indicate what comparisons were made or what hypotheses were being tested. Providing this context would greatly enhance the interpretability and rigor of the findings.

-Please structure the Methods section with appropriate subheadings for clarity.

Results

-Do you mean by RMANOVA, "Repeated Measures Analysis of Variance"? Please clarify.

-The limited sample size of intervention phase (n = 16) raises concerns about the statistical power and generalizability of the study's findings!!! In the discussion section, please address how the small sample size may have influenced your findings and clarify how this limitation should inform the interpretation of your results.

-In qualitative research, sample size is typically determined by the principle of theoretical saturation. However, in this study, the inclusion of 16 participants appears to have been driven primarily by the number of individuals involved in the intervention phase, rather than by saturation criteria.

-The alignment between the interview questions and the reported findings requires further clarification. Specifically, the questioning guide includes prompts about both positive and negative aspects of the program (e.g., perceived barriers, potential negative contributions from newly-joined participants). However, the extracted themes presented in the results section appear to focus exclusively on positive outcomes. This raises concerns about whether the analysis fully captured the range of participant responses or if negative perspectives were underrepresented or omitted. Clarify whether any negative or critical responses were provided by participants and, if so, explain how these were handled during the thematic analysis. If no negative themes emerged, consider discussing this in the limitations or discussion section, including possible reasons (e.g., social desirability bias, group dynamics). Ensure that the themes accurately reflect the scope of the questions asked and the diversity of participant perspectives.

-The final section of the results includes two parts—'Engagement and Expansion' and 'Challenges and Adjustments.' It is unclear whether these are intended as results of the study, and if so, whether they are based on quantitative or qualitative data. Since these sections primarily describe the intervention process and group conditions, their placement within the results may be confusing to readers. It may be more appropriate to relocate this content to the methodology or discussion sections, where it can be contextualized more effectively.

Discussion

-While the effectiveness of interventions in reducing social isolation among older adults has been well-documented, the distinctive strength of the present study lies in the sustained impact of the intervention observed over a two-year period. This long-term effectiveness is a significant contribution and should be emphasized consistently throughout the manuscript, particularly in the discussion section.

-The discussion section would benefit from a more thorough and transparent acknowledgment of the study’s methodological limitations. Currently, the limitations are either briefly mentioned or not addressed in sufficient depth. Given the nature of the study, several methodological constraints—such as small sample size, lack of control group, potential biases, limited generalizability—should be explicitly discussed.

**Do you want your identity to be public for this peer review?** For information about this choice, including consent withdrawal, please see our Privacy Policy

Reviewer #1: No

Reviewer #2: No

---

## [Author Response · Author response to Decision Letter 1]

15 Aug 2025

We would like to confirm that we have responded to all of the reviewers’ comments.

Response to Reviewer 1

Introduction

1- Consider breaking this introduction into shorter paragraphs to improve readability and emphasize key ideas, such as the health risks of loneliness, societal trends, and intervention strategies.

Thank you for your comment. We have revised the introduction section. Please refer to page 3, lines 68-74.

2- The number of citations in the first paragraph is quite dense. Consider integrating the references more smoothly to avoid overwhelming the reader and to improve narrative flow.

Thank you for your comment. We have reduced the number of citations in the first paragraph. Please refer to page 3, lines 62-74.

3- Strengthen the transition between global trends and the specific case of Iran. For example, a sentence explicitly linking the global phenomenon to Iran’s demographic shifts would improve the logical continuity.

Thank you for your comment. It has been revised. Please refer to page 3, lines 68-70.

4- While the background is comprehensive, clarify earlier why a long-term follow-up of the HoSIP program is necessary and what gap it addresses in current literature.

Thank you for your comment. The reason of a long-term follow-up is stated in the last paragraph. Please refer to page 4, lines 117-124; page 5, lines 125-129.

5- When referencing "various interventions" globally, it would be helpful to briefly name a few types (e.g., befriending programs, telehealth-based support) to give readers a clearer idea of what has been tried.

Thank you for your comment. We have revised it. Please refer to page 4, lines 94-95.

6- The final paragraph introduces your study, but the specific research question or hypothesis is not fully clear. Consider adding a sentence like: “This study aims to evaluate whether the effects of the HoSIP program are sustained two years after completion.”

Thank you for your comment. The aim of the study is stated in the last paragraph. Please refer to page 4, lines 117-119.

7- The phrase “older adults” is consistently used, which is good. Ensure the same terminology is used throughout the paper and avoid switching to "elderly" or “seniors” unless quoting from other sources.

Thank you for your comment. The terms “elderly” and “seniors” have been corrected throughout the manuscript. However, the phrase “elderly” in the name of the questionnaire (“the 17-item Self-care Ability Scale for the Elderly”) is part of its official title and therefore has not been changed. Please refer to page 11, line 523; page 12, line 260.

Method

1- The phrase “This mixed method study with a concurrent nested embedded design” is technically accurate but could be clarified for broader readability. Consider briefly describing what is "embedded" in what — e.g., “a primarily quantitative study with a qualitative component embedded for deeper exploration.”

Thank you for your comment. We have revised the sentence. Please refer to page 6, lines 135-136.

2- The recruitment period is described with the exact dates. Consider aligning this level of specificity with the rest of the timeline (e.g., start and end dates of the intervention) for consistency and clarity.

Thank you for your comment. We have revised the section. Please refer to page 6, lines 138-150.

3- The sentence “16 participants (Of the 36 lonely older adults...)” needs refinement in grammar and structure. Consider rephrasing as: “Out of the 36 older adults who originally participated in HoSIP, 16 remained in the follow-up phase and were contacted for this study.”

Thank you for your comment. We have revised the sentence. Please refer to page 8, lines 205-208.

4- The long list of scales makes the paragraph heavy. You could introduce them with a sentence like: “Validated instruments were used to measure key variables, including loneliness, social support, psychological well-being, and self-care ability.” Then list them in smoother format, possibly in a bullet point or table format in the appendix.

Thank you for your comment. We have revised the sentence. Please refer to page 6, lines 158-160; page 7, lines 161-166.

5- Consider briefly explaining how the interview guide was developed. Was it based on previous literature, theoretical models, or pilot testing?

Thank you for your comment. We have revised the sentence. Please refer to page 7, lines 174-180.

6- You could clarify how categories and subcategories were developed — was it inductive or deductive content analysis? That would help readers understand your analytic strategy.

Thank you for your insightful comment. We have revised the sentence. Please refer to page 7, lines 192-193; page 8, lines 194-195.

7- Mentioning COREQ is excellent. Consider explicitly stating that the checklist guided both the design and reporting of the qualitative component for transparency.

Thank you for your suggestion regarding the COREQ checklist. we have explicitly stated in the manuscript. Please refer to page 7, lines 186-188.

Discussion

1- The opening sentence of the Discussion is long and slightly unclear. Consider breaking it into two: one describing the HoSIP program and another summarizing its long-term effects more clearly.

Thank you for your comment. We have revised the sentence. Please refer to page 17, lines 408-426.

2- Good integration of functional theory and digital aging. Consider making the theoretical linkages even more explicit — e.g., how does this support or extend the theory, not just echo it?

Thank you for your comment. We have revised the sentence in the discussion section. Please refer to page 17, lines 416-418.

3- The paragraph discussing quality of life and self-confidence would be stronger if you clearly stated the transition: e.g., “In addition to social benefits, psychological improvements such as self-confidence were also reported.”

Thank you for your comment. We have revised the quality-of-life paragraph. Please refer to page 17, lines 427-431.

4- In the paragraph starting “These findings provide the following insights…”, consider numbering or bulleting the insights (1, 2, 3, etc.) for clarity and easier reading. Each insight is important but gets lost in the long paragraph.

Thank you for your suggestion. We have revised the paragraph starting with “These findings provide the following insights…” by numbering each insight (1, 2, 3, etc.) for greater clarity and readability. Please refer to page 19, lines 491-504.

5- The Conclusion nicely summarizes the findings. You could enhance it by adding: “This model could serve as a blueprint for aging policies and community health strategies in other middle-income countries.”

Thank you for your valuable comment. We have incorporated your suggestion by adding the sentence to the end of the conclusion section. Please refer to page 20, lines 518-519.

Response to Reviewer 2:

Introduction

1- This manuscript aims to evaluate the long-term effects of the HoSIP intervention and serves as a follow-up to the authors’ previous publication. The purpose of the study is not well stated. Was the study aimed at assessing the sustainability of the intervention after it was discontinued after two years? Or is the intervention still ongoing?

Thank you for your valuable comment. We have revised the title “Evaluating the Sustainability and Long-Term Outcomes of the Home Care Support Intervention Program (HoSIP) to Reduce Loneliness among Community-dwelling Older Adults: A two-year follow-up study”. Please refer to page 1, Line 1; page 2, lines 36-38.

Method

2- While the manuscript is well-referenced, the Methods section lacks sufficient detail. In particular, the description of the intervention is unclear. As, mentioned, it is not specified whether the intervention is ongoing or if it was limited to the initial 12-week period. Clarification is needed on whether this study is assessing the sustainability of the intervention's effects two years after its conclusion, or if the intervention has continued in some form during that time. Since the stated aim of the study is to examine the sustainability of the intervention, a more explicit explanation of the timeline and implementation status is essential.

Thank you for your valuable comment. We have revised Methods section. Please refer page 6, lines 138-150.

2- Additionally, the data analysis section requires further elaboration. Although the Results section lists the statistical tests used, it does not clearly indicate what comparisons were made or what hypotheses were being tested. Providing this context would greatly enhance the interpretability and rigor of the findings.

Thank you for your valuable comment. We have revised this section. Please refer page 7, lines 167-172.

3- Please structure the Methods section with appropriate subheadings for clarity.

Thank you for your valuable comment. We have structured the method and material section. Please refer page 6, lines 134, 137, and 151.

Results

1- Do you mean by RMANOVA, "Repeated Measures Analysis of Variance"? Please clarify.

Thank you for your comment. Yes, Repeated Measures Analysis of Variance (RMANOVA). We have added it to the result section. Please refer page 9, line 218.

2- The limited sample size of intervention phase (n = 16) raises concerns about the statistical power and generalizability of the study's findings!!! In the discussion section, please address how the small sample size may have influenced your findings and clarify how this limitation should inform the interpretation of your results.

Thank you for your insightful comments regarding the limited sample size in the intervention phase. please refer to page 19, lines 471-481.

3-In qualitative research, sample size is typically determined by the principle of theoretical saturation. However, in this study, the inclusion of 16 participants appears to have been driven primarily by the number of individuals involved in the intervention phase, rather than by saturation criteria.

Thank you for your comment. We have revised this section in the manuscript. please refer to page 6, lines 152-156.

4-The alignment between the interview questions and the reported findings requires further clarification. Specifically, the questioning guide includes prompts about both positive and negative aspects of the program (e.g., perceived barriers, potential negative contributions from newly joined participants). However, the extracted themes presented in the results section appear to focus exclusively on positive outcomes. This raises concerns about whether the analysis fully captured the range of participant responses or if negative perspectives were underrepresented or omitted. Clarify whether any negative or critical responses were provided by participants and, if so, explain how these were handled during the thematic analysis. If no negative themes emerged, consider discussing this in the limitations or discussion section, including possible reasons (e.g., social desirability bias, group dynamics). Ensure that the themes accurately reflect the scope of the questions asked and the diversity of participant perspectives.

Thank you for highlighting the need to clarify the alignment between our interview questions and reported findings. While our interview guide specifically included prompts addressing both positive and negative aspects of the HoSIP program—such as perceived barriers and potential negative influences from newly joined participants—none of the participants expressed significant negative points regarding the long-term effects or sustainability of HoSIP.

We acknowledge that the absence of negative themes may be influenced by factors such as social desirability bias, group dynamics, or the voluntary nature of participation, which may have led participants to emphasize positive experiences. We have now addressed this issue in the Discussion section as a limitation, and we confirm that our coding and analysis process systematically considered all participant responses, including any critical or negative feedback. please refer to page 19, lines 485-490.

5-The final section of the results includes two parts—'Engagement and Expansion' and 'Challenges and Adjustments.' It is unclear whether these are intended as results of the study, and if so, whether they are based on quantitative or qualitative data. Since these sections primarily describe the intervention process and group conditions, their placement within the results may be confusing to readers. It may be more appropriate to relocate this content to the methodology or discussion sections, where it can be contextualized more effectively.

Thank you for your comment. These parts were not related to quantitative or qualitative results. It just based on the researcher ‘s observation. We have embedded this section in the discussion section. Please refer to page 18, lines 440-442; lines 443-454; lines 461-470.

Discussion

1- While the effectiveness of interventions in reducing social isolation among older adults has been well-documented, the distinctive strength of the present study lies in the sustained impact of the intervention observed over a two-year period. This long-term effectiveness is a significant contribution and should be emphasized consistently throughout the manuscript, particularly in the discussion section.

Thank you for your comments. We have revised the discussion section to highlight the long-term effectiveness and sustainability observed over the two-year follow-up period. Please refer to page 17, lines 480-412; lines 425-426.

2- The discussion section would benefit from a more thorough and transparent acknowledgment of the study’s methodological limitations. Currently, the limitations are either briefly mentioned or not addressed in sufficient depth. Given the nature of the study, several methodological constraints—such as small sample size, lack of control group, potential biases, limited generalizability—should be explicitly discussed.

Thank you for your comments. We have revised the limitation section. please refer to page 19, lines 471-490.

---

## [Editor Report · Decision Letter 1]

25 Aug 2025

Evaluating the Sustainability and Long-Term Outcomes of the Home Care Support Intervention Program (HoSIP) to Reduce Loneliness among Community-dwelling Older Adults:  A two-year follow-up study

PONE-D-25-08918R1

Dear Dr. Nadrian,

We’re pleased to inform you that your manuscript has been judged scientifically suitable for publication and will be formally accepted for publication once it meets all outstanding technical requirements.

Kind regards,

Mehdi Rezaei

Academic Editor

PLOS ONE
---

## [Editor Report · Acceptance letter]

PONE-D-25-08918R1

PLOS ONE

Dear Dr. Nadrian,

I'm pleased to inform you that your manuscript has been deemed suitable for publication in PLOS ONE. Congratulations! Your manuscript is now being handed over to our production team.

Kind regards,

on behalf of

Dr. Mehdi Rezaei

Academic Editor

PLOS ONE